# Functional Attributes of Synovial Fluid from Osteoarthritic Knee Exacerbate Cellular Inflammation and Metabolic Stress, and Fosters Monocyte to Macrophage Differentiation

**DOI:** 10.3390/biomedicines13040878

**Published:** 2025-04-04

**Authors:** Vanshika Srivastava, Abhay Harsulkar, Shama Aphale, Aare Märtson, Sulev Kõks, Priya Kulkarni, Shantanu Deshpande

**Affiliations:** 1Department of Pharmaceutical Biotechnology, Poona College of Pharmacy, Bharati Vidyapeeth Deemed to be University, Erandwane, Pune 411038, India; srivansh2408@gmail.com (V.S.); abhay.harsulkar@bharatividyapeeth.edu (A.H.); shama.aphale@bharatividyapeeth.edu (S.A.); 2Department of Traumatology and Orthopaedics, Institute of Clinical Medicine, University of Tartu, L Puusepa 8, 51014 Tartu, Estonia; aare.martson@kliinikum.ee; 3Clinic of Traumatology and Orthopaedics, Tartu University Hospital, L Puusepa 8, 51014 Tartu, Estonia; 4Perron Institute for Neurological and Translational Science, Nedlands, WA 6009, Australia; sulev.koks@uwa.edu.au; 5Centre for Molecular Medicine and Innovative Therapeutics, Murdoch University, Murdoch, WA 6150, Australia; 6J. Crayton Pruitt Family Department of Biomedical Engineering, University of Florida, 1275 Center Drive JG56, P.O. Box 116131, Gainesville, FL 32611, USA; 7Department of Orthopaedics, Bharati Hospital, Pune-Satara Road, Pune 411043, India

**Keywords:** DAMPs, mitochondrial membrane potential, osteoarthritis, synovial fluid, transcription-factors

## Abstract

**Background:** Besides conventional norms that recognize synovial fluid (SF) as a joint lubricant, nutritional channel, and a diagnostic tool in knee osteoarthritis (kOA), based on the authors previous studies, this study aims to define functional role of SF in kOA. **Methods:** U937, a monocytic, human myeloid cell line, was induced with progressive grades of kOA SF, and the induction response was assessed on various pro-inflammatory parameters. This ‘SF challenge test model’ was further extended to determine the impact of SF on U937 differentiation using macrophage-specific markers and associated transcription factor genes. Mitochondrial membrane potential changes in SF-treated cells were evaluated with fluorescent JC-1 probe. **Results:** a significant increase in nitric oxide, matrix metalloproteinase (*MMP*) 1, 13, and vascular endothelial growth factor (*VEGF*)-1 was noted in the induced cells. A marked increase was seen in *CD68*, *CD86*, and the transcription factors –activator protein *(AP)-1*, interferon regulatory factor *(IRF)-1*, and signal transducer and activator of transcription *(STAT)-6* in the SF-treated cells indicating active monocytes to macrophage differentiation. Reduced mitochondrial membrane potential was reflected by a reduced red-to-green ratio in JC-1 staining. **Conclusions:** these results underline the active role of OA SF in stimulating and maintaining inflammation in joint cells, fostering monocyte differentiation into pro-inflammatory macrophages. The decline in the membrane potential suggestive of additional inflammatory pathway in OA via the release of pro-apoptotic factors and damaged associated molecular patterns (DAMPs) within the cells. Overall, biochemical modulation of SF warrants a potential approach to intervene inflammatory cascade in OA and mitigate its progression.

## 1. Introduction

Knee osteoarthritis (kOA) is the most common form of arthritis in the elderly population across the globe with strong involvement of chronic low-grade inflammation, contributing to radiographic and symptomatic progression of the disease. Synovium and synovial fluid (SF) are the two major sights of inflammation in kOA. Synovium is well connected to the systemic circulation and can effectively modulate the systemic insults into kOA-specific pathological events. In the knee joint, SF is a local counterpart of synovium and a medium of transport across all the joint tissues. This fluid is known to hold pathological signals throughout the kOA progression. Therefore, the synovium-SF axis is pivotal in regulating cartilage loss in kOA [1,2].

Traditionally, SF is recognized as a medium of transport for nutrients across the joint tissues and as a lubricant that ensures frictionless movements. In kOA, SF analysis has been particularly focused as a diagnostic tool in search of biomarkers for accurate staging [3,4] and better assessment of the disease prognosis [5,6,7]. Most of these studies report kOA SF as a ‘biological cocktail’ of growth factors, proinflammatory mediators, extracellular matrix (ECM) proteins, cartilage degradation products, and related proteases. This biological cocktail can induce cellular inflammation as shown by the authors in a previous study [8]. Additionally, SF from other arthropathies has been shown to stimulate metabolic activity in a variety of joint cells including SF-isolated cells [9], ligament-isolated cells [10], and osteoblasts [11]. Considering these observations and the connectivity of SF across the joint tissues, there is a justified potential to predict a much more elaborate functional role for SF in kOA pathology beyond merely serving as a transport medium and as a lubricant.

This study aims to define the functional role of SF in kOA pathology. Knowing that immune cell differentiation into pro-inflammatory phenotype and resultant joint inflammation are directly linked to OA progression [1,12], we investigated the functional involvement of SF by assessing its potential to induce cellular inflammation. The authors additionally hypothesize that SF in OA joints act as a ‘niche’, a physical scaffold that is required to anchor immune cells with necessary microenvironment and survival factors to acquire their functional status. They plan to evaluate the concept of ‘niche’ for SF through its potential to modulate transcription factors that consequentially facilitate the polarization of immune cells into proinflammatory phenotypes.

To achieve this, the authors designed two in vitro models, the SF-challenge test and an in vitro cell differentiation assay. In these models, human monocytes, the primary precursor of synovial inflammation in the disease, were exposed to progressive grades of SF samples obtained from kOA patients. The cellular response was estimated by using different inflammatory and immunological parameters. Further, we explore the impact of SF induction on mitochondrial membrane potential, which is crucially involved in OA development and progression [13]. The cumulative results of these studies helped the authors to define the functional participation of SF in OA pathology.

## 2. Methods

This study abided by the Declaration of Helsinki and all the protocols used in the present communication were approved by the institutional ethics committee (approval number: BVDUMC/IEC/44).

### 2.1. Cell Line

The human monocyte cell line (U937) was purchased from the National Centre for Cell Science (NCCS), Pune, India, and the cells were maintained in Roswell Park Memorial Institute (RPMI)-1640 medium (HiMedia Laboratories, LLC, Kennett Square, PA, USA), 10% Fetal Bovine Serum (FBS) (HiMedia Laboratories, LLC, Kennett Square, PA, USA), 2 mmol/L L-glutamine, 1% penicillin and streptomycin (Sigma-Aldrich, St. Louis, MO, USA), at 5% CO_2_ at 37 °C and 95% relative humidity.

### 2.2. SF Collection

SF samples used for cell induction during the SF-challenge test and the in vitro cell differentiation assay were collected from kOA patients with a confirmed disease diagnosis (age group—40 to 75 years, ethnicity—Indian Asian) at Bharati Hospital and Research Center, Bharati Vidyapeeth University, Pune, India. kOA diagnosis was performed by an experienced orthopedic clinician based on clinical signs, symptoms, and radiographic features. To determine kOA severity, radiographic Kellgren–Lawrence (KL) score system was used. As per this scoring system, grade-I is doubtful narrowing of the joint space and possible, indistinguishable osteophyte; grade-II is definite clearly identifiable osteophytes and possible narrowing of the joint space; grade-III is with moderate multiple osteophytes with definite joint space narrowing, and grade-IV is marked with large osteophytes with significant narrowing of joint space [14]. SF samples from the patients with rheumatoid arthritis (RA), infected arthritis, kOA with tumors, kOA with a history of joint injuries in past 2 years were excluded. All the recruited patients signed an informed written consent form voluntarily before participating in the study.

In total, 36 (*n* = 36; male—23 and female—11) kOA SF samples spanning different KL grades were collected by knee arthrocentesis, following a protocol as reported earlier [15]. SF collection from KL grade-I, II, and III kOA patients was performed only when a patient had a joint effusion and was required to undergo the procedure for symptomatic relief. This procedure took place in a minor operation theatre under strict aseptic conditions. The samples from KL grade-IV kOA patients were collected at the time of knee replacement surgery. SF sample grading was the same as kOA patient grading. For a fair comparison of the selected biomarkers, SF samples with comparable protein levels were included for experimental work in this study. SF protein estimation was performed with standard Bradford assay (Bradford reagent, Sigma-Aldrich, St. Louis, MO, USA).

### 2.3. Determination of kOA SF Potential to Induce Cellular Inflammation

#### 2.3.1. SF Challenge Test

In this assay, U937 cells were exposed to SF samples of progressive KL grades I, II, III, and IV, and the inflammatory response of the induced cells was estimated in terms of nitric oxide (NO) released in cell culture media and gene expression modulation of proinflammatory markers like matrix metalloproteinase-1 (*MMP-1*), *MMP-13* and vascular endothelial growth factor-1 (*VEGF-1*). Details of this cell model are described as follows.

#### 2.3.2. NO Estimation

U937 cells were seeded with a density of 1 × 10^5^ cells/mL in 24-well plates and were induced with 20% (of culture media) SF samples of KL grade-I, II, III, and IV for 48 h. The SF induction concentration was determined based on the results of the cell viability assay, which was performed as described earlier [12,16]. For this experiment, 5 SF samples of each KL grade were used (in total, 5 × 4 KL grades = 20 SFs), and the experiment was performed in triplicates. After 48 h of the incubation period, NO estimation was performed by the Griess reaction method as elaborately described elsewhere [17]. In short, the cell culture media was collected and diluted 1:1 with the solution containing 1% sulfanilic acid and 0.1% N,N-naphthylethylenediamine-dihydrochloride, prepared in 5% phosphoric acid. This mixture was further incubated for 10 min and absorbance was measured at 540 nm. Further calculations were performed in contrast with the standard curve of linear concentration of sodium nitrate, which was used as a standard.

### 2.4. qRT-PCR Analysis of Proinflammatory Markers

qRT-PCR analysis was performed on the U937 cells induced with SF samples of KL grade-II, III, and IV. After 48 h of the induction, the cells were employed for RNA isolation first using TRIzol reagent and later PureLink RNA Mini Kit (Invitrogen Co., Carlsbad, CA, USA) by following the manufacturer’s instructions. Total RNA quantification was done by measurement of UV absorbance at 260 nm. This RNA was further used for cDNA synthesis by using SuperScript-III Cell to cDNA kit (Invitrogen Co., Carlsbad, CA, USA) by following the manufacturer’s instructions. qRT-PCR analysis was performed with Applied Biosystems StepOne Real-Time PCR System using the TaqMan gene expression assays of *MMP-1* (gene assay number—Hs00899658_m1), *MMP-13* (gene assay number—Hs0023392_m1), *VEGF-1* (gene assay number—Hs00900055_m1), and TaqMan Gene Expression Master Mix (Applied Biosystems, Foster City, CA, USA). mRNA levels of all the genes were evaluated against *ACTB* (gene assay number—Hs01060665_m1) as a house-keeping gene using Step One Software version 2.2.2. For the qRT-PCR analysis, 3 SF samples of each KL grade-II, III, and IV were used (total 3 × 3 = 9). The experiment was performed in triplicates and repeated 3 times.

### 2.5. kOA SF Effect on Macrophages Differentiation

#### In Vitro Cell Differentiation Assay

The assay was designed to explore the potential of kOA SF to modulate transcription factor expressions and to drive the immune cell differentiation process. In terms of the experimental procedure, it was a continuation of the SF challenge test. This means we employed the methodology of the SF challenge test, wherein U937 cells were considered as immune cell precursors and exposed to SF samples of progressive KL grades for 48 h. At the end of the induction period, transcription factor modulation was assessed using activator protein-1 (*AP1*), interferon regulator factor-1 (*IRF1*), signal transducer and activator of transcription-1 (*STAT1*), and *STAT6* gene expression, while the status of the newly differentiated cells was assessed using macrophage-specific markers like *CD68*, *CD86*, and *CD163*. The detailed protocol followed for this assay is as follows.

U937 cells were seeded with a density of 1 × 10^6^ cells/mL in a 24-well plate and induced with 20% (of culture medium) of kOA SFs of progressive KL grades for 48 h. We used 9 SF samples of each KL grade-II, III, and IV for the induction. In total, 27 OA SF samples were used. This experiment was performed in triplicates. The cells induced with phorbol 12-myristate 13-acetate (PMA) (Sigma-Aldrich, St. Louis, MO, USA) (dose—100 ng/mL) were used as a positive control and untreated cells were used as a negative control. A safe dose of PMA induction was determined by cell viability assay, which was performed as described elsewhere [16]. To determine a phenotypic shift in the SF-induced U937 cells, the adherent cells were harvested after the induction period (48 h) by treating them with 0.5 mM ethylenediamine tetra-acetic acid (EDTA) in 1× PBS for 15 min at 37 °C and 5% CO_2_ and later dislodged from the bottom of the 24-well plates by vigorous pipetting. These cells were further used for RNA isolation followed by qRT-PCR analysis using the transcription factors (gene assay number—*AP1*-Hs00759776_s1, *IRF1*—Hs05456607_s1, *STAT1*—Hs01013992_g1, *STAT6*—Hs01127463_g1) and macrophage-specific markers (gene assay number—*CD68*—Hs02836816_g1, *CD86*—Hs01567026_m1, *CD163*—Hs00174705_m1), as mentioned earlier in this section.

### 2.6. Functional Analysis

Cytokine/pro-inflammatory marker estimation using the cell culture media was performed as a functional analysis of the in vitro cell differentiation assay in the SF-induced U937 cells after 48 h. In total, 9 SF samples; 3 samples from each KL grade-II, III, and IV, were used in triplicates, and the experiment was repeated 3 times. As the assay did not show significant changes for the M2 phenotype (expression modulation in the CD163 gene), the functional analysis was restricted to pro-inflammatory cytokines, representing the M1 phenotype. Cytokine/pro-inflammatory marker estimation was performed using Human Procartaplex Mix & Match 7-plex (ThermoFisher Scientific, Vienna, Austria; catalog No. PPX-07-MXAADCX) as per the manufacturer’s instruction. For each cytokine, median fluorescence intensity (MFI) was measured by Attune™ NxT Flow Cytometer (ThermoFisher Scientific, Waltham, MA, USA; catalog no. A24863); graphs were plotted using a standard curve.

### 2.7. Determination of SF’s Capacity to Impact Mitochondrial Membrane Potential in kOA

A change in mitochondrial membrane potential (∆ψm) of kOA SF-induced U937 cells was determined in terms of the percentage of the cells with impaired mitochondrial activity. For this, a mitochondrial membrane potential assay kit (with JC-1) (Elabscience, Houston, TX, USA; catalog No.—E-CK-A301) was used as per the manufacturer’s instructions. JC-1 is a fluorescent probe that gives a red fluorescence when it becomes aggregated in the mitochondrial matrix and forms a polymer. Alternatively, in case of a decline in the mitochondrial potential, JC-1 fails to polymerize and produces green fluorescence. Thus, a relative ratio of red and green fluorescence acts as an indicator of impaired mitochondrial membrane potential that may further lead to apoptosis.

In the assay, U937 cells were seeded with 1 × 10^6^/mL density in a 24-well plate and were induced with 20% (of culture media) of SF of KL grade-II and III (n = 18; 9 SF samples from each KL grade-II and III). The cells induced with tumor necrosis factor-α (TNF-α) (dose: 200 ng/mL) were used as a positive control for this set of experiments, wherein a safe induction dose of TNF-α was determined using cell viability assay as described [16]. After 48 h of induction, the cells were harvested by non-enzymatic digestion method using 0.5 mM EDTA in PBS for 15 min at 37 °C and 5% CO_2_. Later, 20 μL of JC-1 (500×) was diluted with 9 mL of ultrapure water to form JC-1 working solution. The working solution was then vortexed for enhanced mixing and then 1 mL of JC-1 assay buffer (10×) was added to the working solution. JC-1 assay buffer (10×) was further diluted to form 1× JC-1 assay buffer. The positive control was prepared by incubating untreated control cells in 10 μM carbonyl cyanide m-chlorophenylhydrazone (CCCP) for 20 min. The SF-induced U937 cells were incubated with 500 μL of JC-1 working solution at 37 °C for 20 min. After the incubation, the cells were washed using JC-1 assay buffer (1×) and analyzed by Attune™ NxT Flow Cytometer. For each experimental reaction, 10,000 events were acquired. The experiment was performed in triplicates. Flow cytometry data analysis was performed using Attune Nxt software (version 2.1), and the results are expressed as the percent stressed cells.

### 2.8. Statistical Analysis

All cell experiment data and qRT-PCR data in various in vitro models were performed in triplicates, repeated for 3 times, and are presented as mean ± SD. Inter-grade statistical significance was determined by One Way ANOVA followed by Bonferroni’s multiple comparison test of significance using GraphPad Prism software version 8.0.2. The data analysis of cytokine/pro-inflammatory marker estimation was performed using Luminex xMAP INTELLIFLEX (version 4.3) and the final concentration of each cytokine/pro-inflammatory marker was calculated in pg/mL.

## 3. Results

### 3.1. SF Induction Generates Cellular Inflammation in U937 Cells

#### 3.1.1. SF Challenge Test

##### NO Estimation

After SF induction, the highest NO release was estimated in the cells induced with KL grade-II SF samples; the release was significantly higher as compared to control (*p* < 0.001) and SF induction with KL grade-I and IV samples (*p* < 0.05). The SF induction with KL grade-I and III samples also revealed a significantly higher NO release in comparison with control cells (KL grade-I vs. control—*p* < 0.001, KL grade-I vs. control—*p* < 0.01), but not with PMA-induced cells (Figure 1). The intergrade comparison of NO release is depicted in Figure 1.

##### qRT-PCR Analysis of Pro-Inflammatory Markers—*MMP-1*, *MMP-13* and *VEGF-1*

The highest *MMP-1* expression was found in the U937 cells induced with KL grade-III SF samples; the expression level was significantly higher as compared to the control (*p* < 0.001). The induction with KL grade-IV samples also caused a significant elevation in *MMP-1* (*p* < 0.01) (Figure 2A).

*MMP-13* gene expression was significantly higher after the induction with SF samples of KL grade-II, III, and IV as compared to control (KL-II vs. control—*p* < 0.01, KL-III vs. control—*p* < 0.01, KL-IV vs. control—*p* < 0.05). U937 cells induced with KL grade-II and III samples revealed a comparable expression of *MMP-13* (Figure 2B).

A peak of *VEGF-1* gene expression was seen in the cells induced with SF samples of KL grade-III (*p* < 0.01). SF induction with KL grade-IV also caused a marginal insignificant increase in the *VEGF-1* gene expression as compared to the control. SF treatment with KL grade-II samples did not cause any significant change in the expression level (Figure 2C).

### 3.2. kOA SF Extends a Proinflammatory Niche for Macrophage Polarization into M1 Type

#### 3.2.1. Modulation of Transcription Factors

After 48 h of SF induction, *AP1*, *IRF1*, and *STAT1* showed a significant KL grade-wise increase when compared to control (Figure 3A–D). AP1 was significantly elevated in the cells induced with KL grade-II and III SF samples in comparison with control cells and the cells induced with KL grade-IV (control vs. KL grade-II: *p* < 0.001; control vs. KL grade-III: *p* < 0.001; KL grade-II vs. KL grade-IV: *p* < 0.001; KL grade-III vs. KL grade-IV: *p* < 0.01;) (Figure 3A). *IRF1* showed a grade-wise increase in the expression level; the highest expression was noted in the cells induced with K grade-IV samples (Figure 3B). No intergrade difference was found in the expression level of this transcription factor. *STAT1* followed an expression trend like *AP1*, and a significant increase was noted in the cells induced with KL grade-II and III when compared to control cells (control vs. KL grade-II: *p* < 0.001; control vs. KL grade-III: *p* < 0.001). Intergrade difference in the *STAT1* expression is presented in Figure 3C. Surprisingly, the induction with KL grade-II and III SF samples caused a significant rise in *STAT6* expression level when compared to control as well as PMA-induced cells (control vs. KL grade-II: *p* < 0.001; control vs. KL grade-III: *p* < 0.001; PMA vs. KL grade-II: *p* < 0.001; PMA vs. KL grade-III: *p* < 0.001). Induction with KL grade-IV samples showed a significant drop in the *STAT6* levels (KL grade-II vs. KL grade-IV: *p* < 0.001; KL grade-III vs. KL grade-IV: *p* < 0.01) (Figure 3D).

#### 3.2.2. Modulation of *CD68*, *CD86* and *CD163* Expression

After SF induction for 48 h, *CD68* expression level was significantly higher in the cells induced with KL grade-II, III, and IV SF samples as compared to the control (control vs. KL grade-II: *p* ˂ 0.01; control vs. KL grade-III: *p* ˂ 0.05; control vs. KL grade-IV: *p* ˂ 0.05). The highest level of this marker was found in the cells induced with KL grade-II SF samples (Figure 4A).

Like *CD68*, *CD86* expression showed a grade-wise decline after SF incubation for 48 h. The highest expression of *CD86* was found in the cells induced with KL grade-II SF samples (*p* ˂ 0.001). The induction with KL grade-III samples also caused a significant elevation in the *CD86* expression level as compared to the control (*p* ˂ 0.05) (Figure 4B). Forty-eight hours of SF incubation did not show any significant modulation of *CD163*. Also, a high dispersion was noticed in the 2^−ΔΔCT^ values of this gene. The highest *CD163* expression level was found in the cells induced with KL grade-II samples. The lowest expression was seen in the cells incubated with KL grade-III SF samples (Figure 4C).

#### 3.2.3. Functional Analysis of SF-Induced U937 Cells

Induction with SF for 48 h showed a significant augmentation in IL-6 release as compared to control as well as PMA induction for all KL grade samples (*p* < 0.01). Also, the induction with KL grade-III and IV samples revealed a significant increase in the IL-6 levels when compared to the induction with KL grade-II SF samples (Figure 5A).

SF induction did not cause any noticeable increase in TNF-α levels when compared to untreated control. Although the release trend of this cytokine matched with NO levels in the SF challenge test, it was not significant on the statistical scale (Figure 1 and Figure 5B).

MMP-13 levels were significantly higher in SF-induced U937 cells when compared to the control (*p* < 0.01); however, they remained lower than the levels in PMA-induced U937 cells. The highest MMP-13 levels were noted in the cells induced with KL grade-III SF samples (Figure 5C).

The highest level of MMP-8 was observed in the cells induced with KL grade-IV SF samples (Figure 5D). The SF-induced cells with KL grade-III and IV showed significantly increased levels in comparison with control and PMA induction as demonstrated in Figure 5D.

### 3.3. kOA SF Causes Reduction in Mitochondrial Membrane Potential Because of Cellular Stress and Inflammation

Notably, the authors did not use PMA-induced U937 cells as a positive control, and chose tumor necrosis factor-α (TNF-α) (dose: 200 ng/mL) a known pre-apoptotic agent [18] as the cell-inducing agent for positive control in this set of experiment. As a known M0 inducer, PMA did not reveal any significant modulation in mitochondrial potential [19,20], and hence was not suitable to consider as a positive control. Interestingly, our decision can be supported by Monteiro et al., 2020, wherein the authors reported an increased mitochondrial membrane potential in M0 cells [20].

Induction with KL grade-II and III kOA SF samples for 48 h revealed a significant increase in the percentage of the cells with impaired mitochondrial activity (*p* ˂ 0.001), confirming a marked decline in mitochondrial membrane potential after induction (Figure 6B). Also, the red-to-green fluorescence ratio of the JC-1 probe, an indicator of mitochondrial depolarization, in the SF-induced cells showed a significant reduction as compared to the untreated cells (Figure 6A,C).

## 4. Discussion

OA is well-received as a low-grade chronic inflammatory disorder [21,22,23]. Inflammation in OA pathology is prominently driven by the synovium–synovial fluid axis of the knee joint [2]. Activated M1 macrophages in the synovium and SF produce several pro-inflammatory markers like IL-1β, IL-6, and TNF-α, which contribute substantially to sustained knee joint inflammation. Therefore, regulatory factors responsible for macrophage polarization can be identified as new targets for OA treatment. In this regard, pathological changes in the OA synovium have been studied elaborately; however, studies on SF are largely focused on its usage as a diagnostic tool. The outlook of the present study was to define the functional role of SF in kOA pathology in terms of its potential to induce cellular inflammation and further provide a pro-inflammatory microenvironment that regulates macrophage polarization. This approach was based on the learning that SF in kOA holds a milieu of pro-inflammatory proteins and can stimulate metabolic changes in a variety of joint cells [8,9,11]. The SF challenge test was designed to test the approach, wherein U937 cells were exposed to progressive grades of kOA SF samples for 48 h. As the cells were provided with the disease-specific micro-environment, we aimed to induce responses that mimic the in vivo conditions observed in OA joints. The cellular response was determined in terms of NO release and gene expressions of pro-inflammatory markers including *MMP-1*, *MMP-13*, and *VEGF-1*. Among these markers, excessive NO is released by activated macrophages under stress as a cytotoxic product and as an indicator of inflammation. Inhibition of TGF-β1, which is crucial in maintaining proteoglycan synthesis and stimulation of MMP production, are major cytotoxic effect of unmitigated NO in the context of OA pathology [24]. On the other hand, *MMP-1*, *MMP-13*, and *VEGF-1* are produced in excess amounts by the inflamed macrophages and serve as indirect markers of inflammation induced in response to the local microenvironment [25]. A significant increase in all these markers after SF induction is therefore a confirmation of inflammation induction and macrophage activation in kOA. Importantly, besides U937 cells, the results of the SF challenge test were consistent on human synoviocytes (SW982) and rat synoviocytes underlining the SF’s unequivocal potential to stimulate cellular inflammation [2,8]. Here, the authors want to acknowledge that patient characteristics like age and body mass index (BMI) are well-known major factors that influence the inflammatory milieu of OA SF serving as a source of systemic inflammation and oxidative stress [26,27]. To neutralize the influence of these factors on the study outcomes, SF samples of comparable age and BMI from KL grade-I, II, III, and IV were selected for the cell induction.

Macrophages are known to respond to their stimuli by modifying phenotype; based on the type of stimulant, effector macrophages are categorized into M1 and M2 phenotypes, which release pro-inflammatory and anti-inflammatory cytokines, respectively [12,28]. The enhanced polarization of the M1 phenotype is crucial in OA progression as it is associated with a release of several proinflammatory factors and subsequent cartilage degradation and osteophyte formation [29]. In vitro cell differentiation assay in the present work was an approach to determine if kOA SF can act as a ‘niche’ through its disease-specific pathobiological milieu for macrophage polarization. A concept of ‘niche’ is postulated, which supplements the cells with a physical scaffold and required microenvironment to facilitate their tissue-specific adaptation process [30,31]. SF induction for 48 h revealed a significant increase in *CD68* and *CD86* expression, particularly for the induction with KL grade-II and III SF samples (Figure 4A,B). The higher dispersion in Figure 4 denotes variation among the SFs in factors that can modulate expression of *CD68* and *CD86* genes further strengthening the ‘niche’ function of SFs. This expression trend corroborates well with inflammation levels revealed in the SF challenge test (Figure 1) and the cytokines estimation performed as a functional analysis of the in vitro cell differentiation assay (Figure 5). The individual role of each selected cytokine is well recognized in the OA pathology as defined elsewhere [32,33] and hence not discussed here. CD163 representing M2 type of macrophages however did not show any significant modulation after SF treatment.

Modulation of transcription factors linked with macrophage polarization is an indirect yet effective way of confirming active participation of SF in the polarization process. This is because transcription factors are the proteins that regulate and perpetuate the expression of many genes including proinflammatory genes [34]. Here, we evaluated expressions of transcription factors like *IRF1*, *STAT1*, *AP1*, and *STAT6* in the SF-induced U937 cells. Among these factors, *IRF1*, *STAT1*, and *AP1* regulate M1 macrophage polarization, while *STAT6* is associated with IL-4 and IL-13-mediated M2 macrophage polarization [35,36]. As shown in Figure 3B, SF induction with KL grade-II, III, and IV samples caused a significant upregulation of *IRF1*. *AP1* and *STAT1* revealed a similar expression trend, although the increase in *AP1* was insignificant on the statistical scale. There are several documented evidence that underline an association between these transcription factors and macrophage polarization. For example, the *AP1* transcription factor family is primarily associated with regulating the production of pro-inflammatory cytokines. Activation of toll-like receptor (TLR) that further induct mitogen-activated protein kinase (MAPK) signaling cascade through MyD88, results in the activation of *AP1* in responding macrophage [37,38]. In the present work, it can be inferred that damage-associated molecular patterns (DAMPs) present in kOA SF can agonize TLR responses and ultimately cause AP1 upregulation in the induced U937 cells. This statement is supported by significantly higher levels of MAPKs as observed in SF proteome analysis performed in one of the authors’ previous studies [12]. *IRF1* can be robustly up-regulated by *IFN-γ* subsequently inducing M1 polarization in the inflammatory microenvironment; in the present context, kOA SF was expected to provide the required inflammatory microenvironment [39,40,41]. Also, a synergistic action between *IRF1* and *IRF8* due to a common IAD2 domain in molecular structure, results in amplified interleukin-12 production revealing another evidence that *IRF1* promotes M1 polarization [42,43].

*STAT1* and *STAT6* are involved in M1 and M2 types of polarization, respectively [44]. *STAT1* mediate interferon-γ (*IFN-γ*) induced M1 polarization; IFN-γ binding to its receptor induces Janus kinase (JAK) ½-mediated tyrosine phosphorylation and subsequent dimerization of *STAT1*, which binds as a homodimer to IFN-γ activated sites on the promoter region of M1-specific genes like *iNOS2* and MHC-II [36]. Also, *IL-6* binding to its receptor IL-6R stimulates gp-130-mediated activation of *STAT1*. To note, a high level of *IFN-γ* and *IL-6* in kOA SF can therefore activate *STAT1* upregulation in the induced cells [45]. Furthermore, a positive correlation between *STAT1* and *CD68* as reported by Kasperkovitz et al., 2003 [46] in an RA synovium biopsy study, is in support of the present study; a comparable expression trend was observed for both markers in U937 cells after SF induction (Figure 3C and Figure 4A). On the other side, *STAT6* is involved in *IL-4* and *IL-13*-mediated M2 macrophage polarization through a common receptor—IL-4 receptor-α. This follows subsequent phosphorylation of *STAT6* by JAK1/JAK3 or JAK1/Tyk2 in the case of IL-4 or IL-13 binding, respectively. Target genes of *STAT6* include *Arg-1*, *Mrc1/CD206*, and resistin-like-α; all these genes are linked with M2-type macrophage polarization [46,47]. *STAT6*-deficient mice model exhibited an augmented production of chemokine (monokine of IFNγ) because of IL-4–mediated suppression of the IFNγ-STAT1 pathway and loss in the IL-4–induced Th2 response, revealing an indispensable role of *STAT6* in IL-4–mediated inhibition of many pro-inflammatory genes [47]. In the current study, although we observed a significant modulation in *STAT6* following the SF induction, it was not proportionally reflected in the *CD163* expressions as its target gene (Figure 4C). This could be attributed to dominance of proinflammatory mediators in kOA SF, which could outweigh the regulatory effect of *STAT6* on *CD163* [45]. Altogether, a significant modulation in M1 regulatory transcription factors like *IRF1* and *AP1* in SF-induced U937 cells suggests that SF plays an active role in driving the polarization of macrophages into a proinflammatory phenotype. These macrophages are functionally viable and thereby contribute to the sustained inflammation in OA joints.

SF serves as a pool for the accumulation of DAMPs during OA progression. SF proteome as well as ELISA analysis performed by the authors in their earlier studies had revealed a variety of differentially expressed proteins such as alarmins, free histones, high-mobility group box proteins (HMGBPs), and other extracellular molecular fragments (ECM) including proteoglycans and glycosaminoglycan that act as DAMPs and stimulate inflammatory response [12,14]. Among these proteins, alarmins, free histones, and HMGBPs act as intracellular DAMPs, while ECM fragments serve as extracellular DAMPs [48,49,50]. The binding of DAMPs to toll-like receptors-2/4 (TLR-2/4), typically activates nuclear factor-κB (NF-κB), an inflammatory response key regulator [51]. It is known that TLR-2/4 facilitates M1 macrophage polarization [52,53]. Inflammatory responses stimulated through the activation of TLR-2/4, also increase the production of MMPs, maintaining a vicious circle of OA pathology. Importantly, these differentially expressed proteins revealed a particular trend of expression as their maximum levels in KL grade-II and III SF samples, which very well correlated with the expression trend of *CD68* and *CD86* in the respectively induced U937 cells. Thus, the results of the SF challenge test in the present work and SF analysis as described in Kulkarni et al. 2016, Ingale et al. 2021 and Kulkarni et al. 2022 clearly suggest progressive accumulation of DAMPs in kOA SF has a substantial contribution in transforming this fluid into a pro-inflammatory microenvironment in OA [2,12,14]. Additionally, kOA SF also holds several other proteins including macrophage migration inhibitory factor (MIF), macrophage capping protein (MCP), and many RAS-related RAB proteins, which are linked with macrophage regulation as discussed in Kulkarni et al. 2022 [12].

Reduced mitochondrial membrane potential is a classic feature of OA chondrocytes [13,54]. A decline in the mitochondrial membrane potential (Δψm) leads to mitochondrial swelling, disruption of its outer membrane, and release of cytochrome C, a pro-apoptotic protein. Therefore, elevated chondrocyte apoptosis, mitochondrial dysfunction, and OA lesions are well correlated [13]. Also, increased NO levels, as found in OA joints, contribute to chondrocyte apoptosis by causing disruption in mitochondrial energy metabolism and increased ROS production [54,55]. Mitochondrial dysfunction in OA is a relatively less presented concept and was investigated in this work. The effect of SF induction on mitochondrial potential in U937 cells was determined using JC-1, a fluorescent probe, which produces a red fluorescence when aggregated in the mitochondrial matrix in the form of a polymer. In case of reduced mitochondrial membrane potential, JC-1 fails to accumulate and, as a monomer, produces green fluorescence. A significant decline in the red/green ratio in the induced U937 cells by all KL grade SF samples was therefore evidence of a strong decline in their mitochondrial membrane potential after induction (Figure 6A,C). Nonetheless, this effect of kOA SF can be an additional source of inflammation in OA joints as mitochondrial DAMPs and the DAMPs from apoptotic cells can trigger immune responses promoting the release of proinflammatory mediators that further induce ROS production and mitochondrial dysfunction, continuing a loop of OA pathology [56,57].

Conclusively, the outcomes of the present work clearly underline a key participation of kOA SF in stimulating and maintaining inflammation in joint cell types. The pro-inflammatory microenvironment (provided by kOA SF) drives immune cell differentiation into a pro-inflammatory phenotype, which is responsible for maintaining a cascade of pathological events in the OA joints. Furthermore, this fluid also exhibits the pro-inflammatory effect by negatively affecting mitochondrial potential as shown on U937 cells in this work. Most of these activities by kOA SF are closely linked to the DAMPs and pro-inflammatory factors (released by synovium, cartilage, and the SF cells). In other words, this fluid prominently acts through DAMPs and pro-inflammatory factors present in it. Consequently, modulation of SF has the potential to halt the progressive inflammation in OA joints.

## Figures and Tables

**Figure 1 biomedicines-13-00878-f001:**
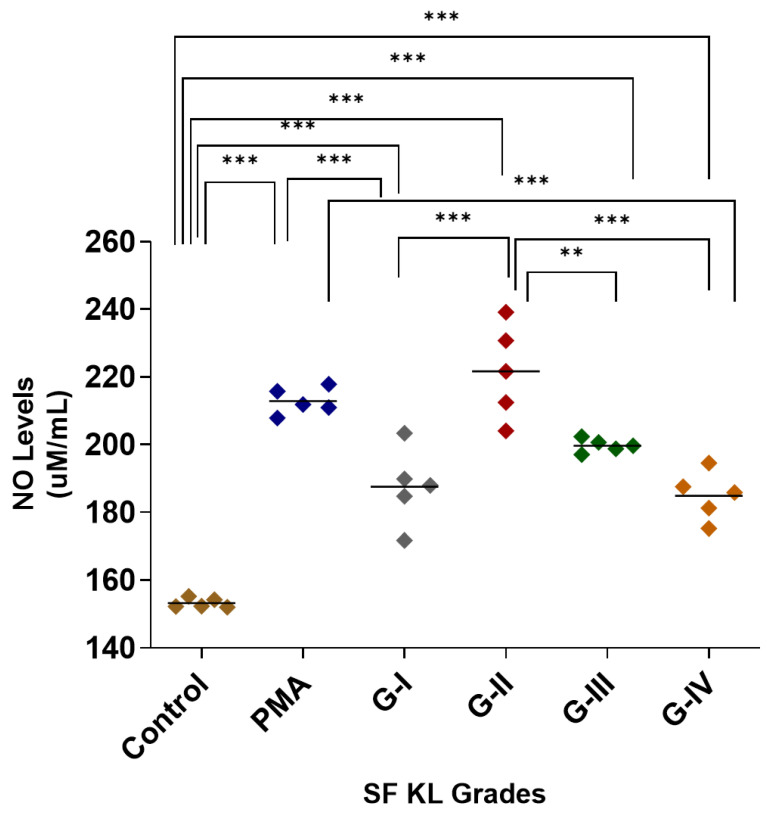
SF challenge test: NO estimation. U937 cells were induced with 20% SFs (of culture media) of progressive KL grades for 48 h, and inflammatory response was estimated in terms of NO released in cell culture media; the cells induced with PMA stood as positive control; the experiment was performed in triplicates and repeated for three times., A total of 5 SF samples per KL grade were used to induce inflammation (n = 5 × 4 = 20). Intergrade comparison with statistical significance is indicated by the lines; for statistical significance—** *p* < 0.01, and *** *p* < 0.001. (KL—Kellgren–Lawrence, NO—nitric oxide, PMA—phorbol 12-myristate 13-acetate, SF—synovial fluid).

**Figure 2 biomedicines-13-00878-f002:**
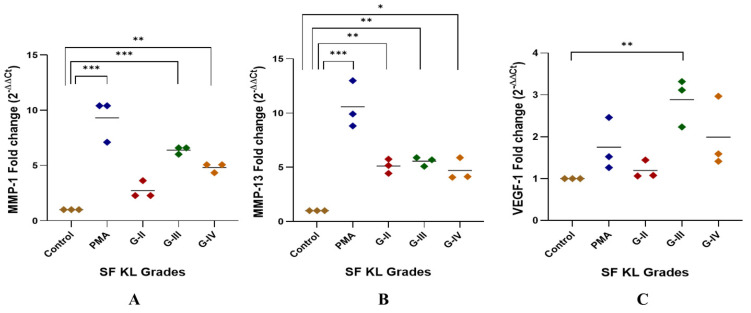
SF challenge test: qRT-PCR analysis of pro-inflammatory markers. SF-induced (20% of culture media) U937 cells were used for RNA isolation followed by cDNA preparation to detect fold change in mRNA abundance of *MMP-1*, *MMP-13*, and *VEGF-1* as presented in the scatter plots (**A**), (**B**), and (**C**), respectively; in all these experiments, *ACTB* was used as a housekeeping gene; for statistical significance—* *p* < 0.05, ** *p* < 0.01, and *** *p* < 0.001 when compared to control.

**Figure 3 biomedicines-13-00878-f003:**
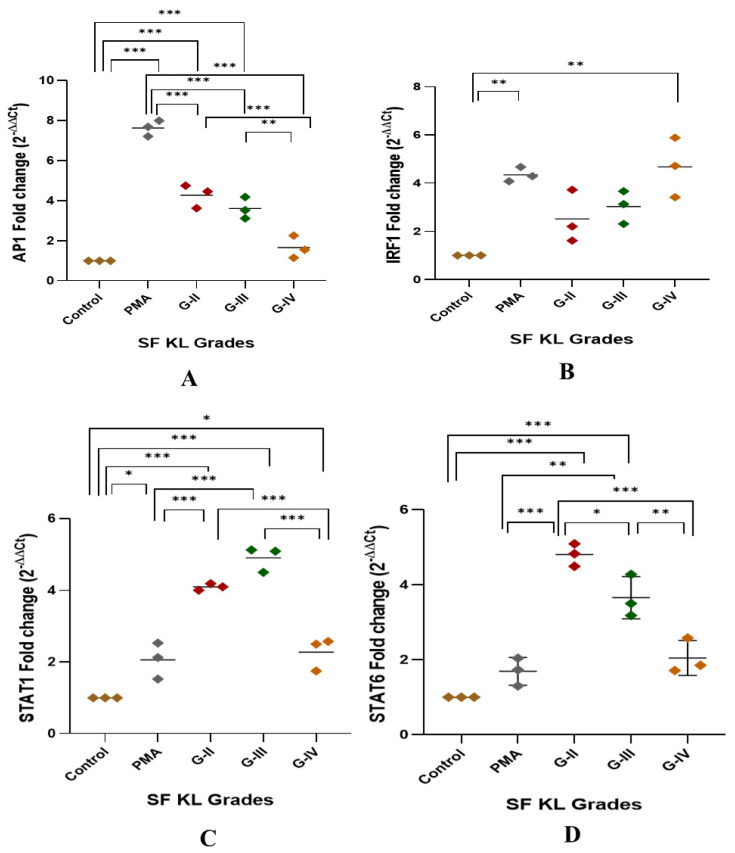
Modulation of transcription factors linked with macrophage polarization after SF induction. U937 cells were exposed to SF samples (20% of culture media) from KL grades—II, III, and IV for 48 h followed by the determination of modulation of *AP1*, *IRF1*, *STAT1*, and *STAT6* to detect fold change in mRNA abundance as depicted in (**A**), (**B**), (**C**), and (**D**), respectively. In this set of experiments, uninduced cells and the cells induced with PMA were used as a control and positive control, respectively. Intergrade comparison for all the transcription factors is indicated by the lines; for statistical significance—* *p* < 0.05, ** *p* < 0.01, and *** *p* < 0.001.

**Figure 4 biomedicines-13-00878-f004:**
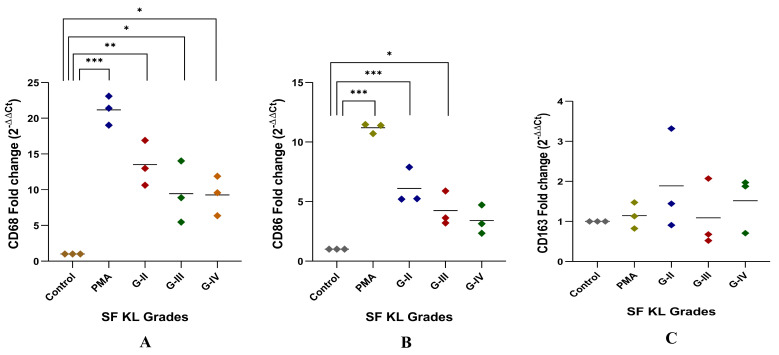
In vitro cell differentiation assay. SF challenge test using samples of KL grade—II, III, and IV were used to determine a phenotype shift in the induced cells by gene assays of *CD68*, *CD86* representing M1 macrophages, and *CD163* representing M2 macrophages as shown in (**A**), (**B**), and (**C**), respectively; for statistical significance—* *p* < 0.05, ** *p* < 0.01, and *** *p* < 0.001 when compared to control cells.

**Figure 5 biomedicines-13-00878-f005:**
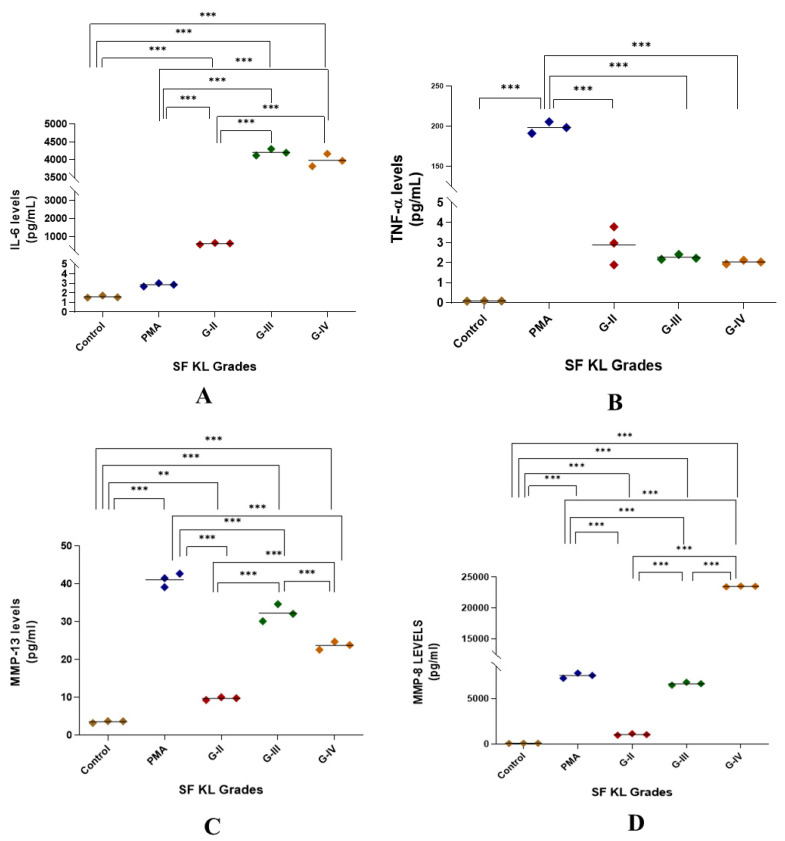
Functional analysis of the SF-induced U937 cells in in vitro cell differentiation assay. After 48 h of SF induction, the cell culture media was used for the estimation of cytokines as a form of a functional analysis of the induced cells; the levels of IL-6, TNF-α, MMP-13, and MMP-8 in the cell culture media are shown in the scatter plots (**A**), (**B**), (**C**), and (**D**), respectively; for statistical significance—** *p* < 0.01, and *** *p* < 0.001 when compared to control.

**Figure 6 biomedicines-13-00878-f006:**
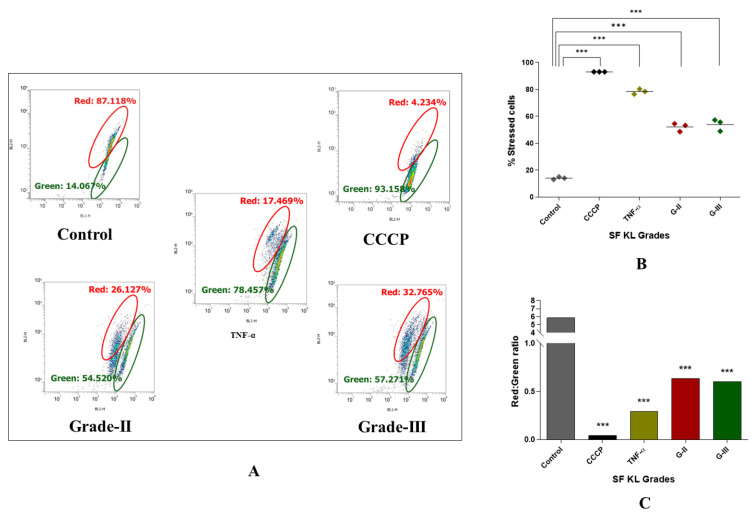
Determination of SF’s capacity to impact mitochondrial membrane potential in kOA. U937 cells were induced with SF samples of KL grades—II and III and further were subjected to flow-cytometry analysis using JC-1 fluorescent probe to determine cellular mitochondrial potential; (**A**) representative scatter plots depicting percentage of the cells with red and green fluorescence by JC-1 probe after the SF induction; here, red fluorescence is indicative of the cells with normal mitochondrial membrane potential (∆ψm) while green fluorescence shows the cells with a reduced mitochondrial membrane potential (∆ψm); (**B**) scatter plot depicting percentage of stressed cells, as an indicative of declined mitochondrial membrane potential after SF induction for 48 h; (**C**) a grade wise estimation of red/green ratio in the induced U937 cells after 48 h; for statistical significance—*** *p* < 0.001, as compared to control cells (CCCP—carbonyl cyanide m-chlorophenyl hydrazone).

## Data Availability

The datasets used and/or analyzed during the current study are available from the corresponding author upon reasonable request, ethics policy limits public access to dat.

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
