# Peer review of "Functional Attributes of Synovial Fluid from Osteoarthritic Knee Exacerbate Cellular Inflammation and Metabolic Stress, and Fosters Monocyte to Macrophage Differentiation"

_biomedicines, 2025, doi:10.3390/biomedicines13040878_

Round 1
Reviewer 1 Report
Comments and Suggestions for Authors
This study investigates how synovial fluid from patients with KOA promotes cell inflammation, oxidative stress, and induces the differentiation of monocytes into macrophages. The findings deepen the understanding of the pathogenesis of osteoarthritis and hold significant scientific value.
In the methods section,the authors used 20% synovial fluid for cell treatment and suggest adding preliminary experimental data or literature to support this choice.
This study collected synovial fluid from 36 KOA patients and grouped them according to KL grading. However, the differences in inflammation levels among patients with KL grades I, II, III, and IV may be influenced by factors such as age, BMI, and gender. It is recommended to discuss these potential confounding factors.
Author Response
Response to the Reviewer 1
This study investigates how synovial fluid from patients with KOA promotes cell inflammation, oxidative stress, and induces the differentiation of monocytes into macrophages. The findings deepen the understanding of the pathogenesis of osteoarthritis and hold significant scientific value.
The authors’ response:
The authors thank the reviewer for his/her appreciation.
Comment 1: In the methods section, the authors used 20% synovial fluid for cell treatment and suggest adding preliminary experimental data or literature to support this choice.
The authors’ response: SF concentration for cell induction (20%) was determined by cell viability assay outcomes (refer lines 130-132). The assay outcomes revealed about 90% cell survival after 48 hours of SF treatment at 20% concentration. Also, the authors have previously published the study, wherein they used the 20% SF induction concentration (Kulkarni, P.; Srivastava, V.; Tootsi, K.; Electricwala, A.; Kharat, A.; Bhonde, R.; Koks, S.; Martson, A.; Harsulkar, A. Synovial Fluid in Knee Osteoarthritis Extends Proinflammatory Niche for Macrophage Polarization. Cells 2022, 11, 4115. https://doi.org/10.3390/cells11244115). This study has been cited in support of the methodology in the manuscript text now.
Comment 2: This study collected synovial fluid from 36 KOA patients and grouped them according to KL grading. However, the differences in inflammation levels among patients with KL grades I, II, III, and IV may be influenced by factors such as age, BMI, and gender. It is recommended to discuss these potential confounding factors.
The authors’ response: The authors appreciate the reviewer for his /her suggestion. The suggested points are now discussed in the manuscript and the following lines are added in the discussion session (lines: 398-403).
Here, the authors want to acknowledge that patient characteristics like age and body mass index (BMI) are well-known major factors that influence the inflammatory milieu of OA SF serving as source of systemic inflammation and oxidative stress (doi: 10.1016/j.joca.2015.01.008; doi: 10.1038/nrrheum.2016.65). To neutralize the influence of these factors on the study outcomes, SF samples of comparable age and BMI from KL grade-I, II, III and IV were selected for the cell induction.
Reviewer 2 Report
Comments and Suggestions for Authors
The article by V. Srivastava et al. investigates the influence of synovial fluid factors in patients with osteoarthritis on the inflammatory status, mitochondrial state of U937 monocytes and their ability to differentiate into macrophages. The study revealed a highly significant effect of synovial fluid on monocytes, leading to increased expression of pro-inflammatory factors, decreased mitochondrial potential and differentiation of monocytes into pro-inflammatory macrophages. The data obtained suggest a significant role for humoral factors contained in the interstitial fluid in the mobilization of inflammatory cells that are a factor in the accelerated development of osteoarthritis.
Despite the considerable work done by the authors, it should be recognized that in its current form the article is not ready for publication and requires substantial revision.
Reviewer’s comments
- The title of this manuscript should be changed because oxidative stress was not studied in this work.
- Lines 283 (section 3.2) and 72: The word 'niche' is mentioned several times by the authors. They have to define it somewhere, preferentially in Introduction.
- Section 3.2.1. and Figure 3 describe the results of determining the expression levels of the transcription factors STAT1 and STAT6 by PCR. It is known that the activity of transcription factors is determined by their phosphorylation and dimerization. To what extent can the transcriptional activity of the genes of these factors be used to assess the activity of the factors themselves?
- Lines 206-216: This text would be better moved (with some modifications) to Chapter 3.3 (lines 373-380) to explain the choice of cell stimulation method (TNF is better than PMA...). It would also be desirable to provide data characterizing the mitochondrial potential measured in the presence of PMA (no change compared to the control) in Figures 6 A-C, in order to give a complete picture of the results obtained.
- Figures 2-6: The figure legend should only describe experimental conditions, statistics and abbreviations. Explanatory sentences (like “a significant increase was noted in the U937 cells induced with KL grade-II SF samples…” and so on) should be transferred into appropriate places in Results or Discussion.
- Figure 4C – why scattering is represented for this graph only?
why the spreads are only on this graph
- Lines 362-364: whether this sentence is necessary?!
Stylistic remarks
- Lines 61-63: … as showed by the authors in their previous study… = as shown by the authors in a previous study
- Line 108: In total, thirty-six (n = 36; male – 23 and female – 11) kOA SF samples… (however, 23+11=34!?)
- Line 142: using TRIzol re[a]gent and later PureLink RNA Mini Kit (Invitrogen Co., Carlsbad CA,
- Line 155: …2.5. kOA SF Provides a Proinflammatory Niche for Immune Cell Differentiation…
- Line 155: “…2.5. kOA SF Provides a Proinflammatory Niche for Immune Cell Differentiation…” Authors need to change that title, to make more appropriate for Materials and Methods (Something like “kOA SF effect on U937 cell differentiation”)
- Line 215: De Brito Monteiro, 2020 – what is the reference number for this?
- Lines 61-63: … as showed by the authors in their previous study… = as shown by the authors in a previous study
- Line 466: Agonize - ?! (to serve as agonists?)
English is readable but needs some improvement. It could be recommended to authors to re-read the manuscript to correct misprints and punctuation
Author Response
Response to the reviewer 2:
Comment 1: The title of this manuscript should be changed because oxidative stress was not studied in this work
The authors’ response: The manuscript title is changed to its new version as: “Functional Attributes of Osteoarthritic Knee Synovial Fluid Exacerbate Cellular Inflammation, Metabolic Stress and Monocyte to Macrophage Differentiation”
Comment 2: Lines 283 (section 3.2) and 72: The word 'niche' is mentioned several times by the authors. They have to define it somewhere, preferentially in Introduction.
The authors’ response:
The authors thank the reviewer for this pointer. The following lines defining the concept of ‘niche’ has been added to introduction for better understanding (Lines: 71-76).
The authors additionally hypothesize that SF in OA joints act as a ‘niche’, a physical scaffold that is required to anchor immune cells with necessary microenvironment and survival factors to acquire their functional status. They plan to evaluate the concept of ‘niche’ for SF through its potential to modulate transcription factors that consequentially facilitates polarization of immune cells into proinflammatory phenotypes.
Comment 3: Section 3.2.1. and Figure 3 describe the results of determining the expression levels of the transcription factors STAT1 and STAT6 by PCR. It is known that the activity of transcription factors is determined by their phosphorylation and dimerization. To what extent can the transcriptional activity of the genes of these factors be used to assess the activity of the factors themselves?
The authors’ response:
This is a very intelligent question. Yes, only qRT PCR is not sufficient to comment on STAT1/STAT6 action as their activity is dependent on phosphorylation and dimerization and there are other factors like SOCS1 that can decisively determine their activation. However, their expression does indicate priming status of cells for the cytokine response. Further, we may like to bring to notice, that IRF1 (Figure 3B) is the target gene of STAT1, its expression status certainly reiterates STAT1 activation and function.
Comment 4: Lines 206-216: This text would be better moved (with some modifications) to Chapter 3.3 (lines 373-380) to explain the choice of cell stimulation method (TNF is better than PMA...). It would also be desirable to provide data characterizing the mitochondrial potential measured in the presence of PMA (no change compared to the control) in Figures 6 A-C, in order to give a complete picture of the results obtained.
The authors’ response:
The authors thank the reviewer for his / her suggestion. The indicated lines (line 206-216) are now removed from the Methods section and shifted to the indicated chapter 3.3 with the modifications as showed below (lines 345-351) –
“Notably, the authors did not use PMA induced U937 cells as a positive control, and chose tumor necrosis factor-α (TNF-α) (dose: 200ng/ml) a known pre-apoptotic agent (19) as the cell inducing agent for positive control in this set of experiment (possible figure numbers). As a known M0 inducer, PMA did not did not reveal any significant modulation in mitochondrial potential (20,21), and hence was not suitable to consider as a positive control. Interestingly, our decision can be supported by Monteiro et al, 2020, wherein the authors reported an increased mitochondrial membrane potential in M0 cells (21).”
Hereby, the authors provide a comparative bar graph of the induction comparison of PMA and TNF- α during the JC-1 assay for the reviewer’s reference. They feel confident that the scientific context of choosing TNF- α over PMA has been sufficiently explained in the manuscript text.
A bar graph revealing a comparative cellular stress induction by PMA and TNF-A in JC-1 assay
Comment 5: Figures 2-6: The figure legend should only describe experimental conditions, statistics and abbreviations. Explanatory sentences (like “a significant increase was noted in the U937 cells induced with KL grade-II SF samples…” and so on) should be transferred into appropriate places in Results or Discussion.
The authors’ response:
Accepting the reviewer’s observation all the Figure ligands are reviewed and explanatory sentences are removed. The figure legends for the figures 2-6 are now modified as per the suggestion of the reviewers (Lines: 260-264, 293-299, 307-311, 326-330, 360-370).
Comment 6: Figure 4C – why scattering is represented for this graph only?
The authors’ response: The authors thank the reviewer for pointing out this issue. This was by mistake and now has been rectified (Figure 4C).
Stylistic remarks
Overall response by the authors: The authors thank the reviewer for his / her stylistic remarks for improving the quality of manuscript. All these suggestions are implemented as indicated by the line numbers below.
Comment 1: Lines 61-63: … as showed by the authors in their previous study… = as shown by the authors in a previous study
The authors’ response: lines 61-63.
Comment 2: Line 108: In total, thirty-six (n = 36; male – 23 and female – 11) kOA SF samples… (however, 23+11=34!?)
The authors’ response: the indicated SF donor number is correct; the collected SF samples include some bilateral SF samples. 2 donors (of KL Grade IV) consented for the collection of bilateral samples (line 111).
Comment 3: Line 142: using TRIzol re[a]gent and later PureLink RNA Mini Kit (Invitrogen Co., Carlsbad CA,
The authors’ response: line 145
Comment 5: Line 155: “…2.5. kOA SF Provides a Proinflammatory Niche for Immune Cell Differentiation…” Authors need to change that title, to make more appropriate for Materials and Methods (Something like “kOA SF effect on U937 cell differentiation”)
The authors’ response: kOA SF effect on macrophages differentiation (line 158).
Comment 6: Line 215: De Brito Monteiro, 2020 – what is the reference number for this?
The authors’ response: The reference number for the above reference is 21. For clarity the citation in the text has been changed to Monteiro et al, 2020 (line 350) and citation number has also been added (line 351).
Comment 8: Line 466: Agonize - ?! (to serve as agonists?)
The authors’ response: Yes, the word “agonize” in line 443 symbolizes the agonistic action of DAMPs on TLRs responses and ultimately causing AP1 upregulation.

Round 2
Reviewer 2 Report
Comments and Suggestions for Authors
The authors of the manuscript under review have fully corrected the text and addressed all the reviewer's comments. Therefore, the manuscript has been sufficiently improved to warrant publication in Biomedicines. This paper may be of interest to doctors and scientists working on the cellular and molecular mechanisms involved in the development of osteoarthritis